# Barriers and Facilitators for the Implementation and Evaluation of Community-Based Interventions to Promote Physical Activity and Healthy Diet: A Mixed Methods Study in Argentina

**DOI:** 10.3390/ijerph16020213

**Published:** 2019-01-14

**Authors:** Maria Belizan, R. Martin Chaparro, Marilina Santero, Natalia Elorriaga, Nadja Kartschmit, Adolfo L. Rubinstein, Vilma E. Irazola

**Affiliations:** 1Instituto de Efectividad Clinica y Sanitaria (IECS), Dr. Emilio Ravignani 2024, C1414CPV Buenos Aires, Argentina; mchaparro@iecs.org.ar (R.M.C.); msantero@iecs.org.ar (M.S.); nelorriaga@iecs.org.ar (N.E.); adolfo.rubinstein@gmail.com (A.L.R.); virazola@iecs.org.ar (V.E.I.); 2Institute of Medical Epidemiology, Biometrics and Informatics, Martin-Luther-University Halle-Wittenberg, 06097 Halle (Saale), Germany; nadja.kartschmit@uk-halle.de

**Keywords:** healthy environments, physical activity, healthy diet, health promotion

## Abstract

*Background*: Obesogenic environments promote sedentary behavior and high dietary energy intake. The objective of the study was to identify barriers and facilitators to the implementation and impact evaluation of projects oriented to promote physical activity and healthy diet at community level. We analyzed experiences of the projects implemented within the Healthy Municipalities and Communities Program (HMCP) in Argentina. *Methods*: A mixed methods approach included (1) in-depth semi-structured interviews, with 44 stakeholders; and (2) electronic survey completed by 206 individuals from 96 municipalities across the country. *Results*: The most important barriers included the lack of: adequate funding (43%); skilled personnel (42%); equipment and material resources (31%); technical support for data management and analysis (20%); training on project designs (12%); political support from local authorities (17%) and acceptance of the proposed intervention by the local community (9%). Facilitators included motivated local leaders, inter-sectorial participation and seizing local resources. Project evaluation was mostly based on process rather than outcome indicators. *Conclusions*: This study contributes to a better understanding of the difficulties in the implementation of community-based intervention projects. Findings may guide stakeholders on how to facilitate local initiatives. There is a need to improve project evaluation strategies by incorporating process, outcome and context specific indicators.

## 1. Introduction

Non-communicable diseases (NCDs) such as cardiovascular diseases (CVDs) and type 2 diabetes (T2D) account for a large part of the global burden of disease and rank highly among the causes of death worldwide [1]. Commonly, unhealthy lifestyle habits, notably physical inactivity and poor diet, are among the main reasons for developing NCDs [2]. In the Southern Cone of Latin America (composed of Argentina, Uruguay and Chile) obesity, physical inactivity and poor diet are among the ten major risk factors according to the assignable burden of disease [3].

While prevention of NCDs has historically and still emphasizes individual health-behaviors, there is an increasing acknowledgment of the influence of social and physical environments in which people live [4,5,6,7,8,9]. Environments that promote sedentary behavior and high dietary energy intake are considered to be “obesogenic” while hindering leisure-time physical activity and active transport. The sum of man-made or modified characteristics of the physical environment, the so-called built environment, represents a focal point of the obesogenic environment, especially in urban areas [10,11,12,13,14,15]. Built environments influence physical activity and diet of the population and, hence, the burden of NCDs [16,17].

The Pan American Health Organization (PAHO) implemented the Healthy Municipalities and Communities Program (HMCP) in 1990 founded on the Ottawa Charter principle of ‘enabling and empowering people to take control over and improve the determinants of health’ [18]. The HMCP places an emphasis on determinants of health and the collaboration among the community, the local government authorities, and other key stakeholders in order to enhance quality of life. The PAHO Guideline (Healthy Municipalities & Communities: Mayors’ Guide for Promoting Quality of life) recommends a set of impact indicators that can be used by Mayors or Community leaders [19]. 

In Argentina, the HMCP Program has been led by the National Ministry of Health (MoH) since 2003. The program provides a policy framework that enables participation and collaboration among community stakeholders, relevant authorities, and local actors including four areas of interest: Health systems and services, lifestyles, environmental health, and socioeconomic factors [20]. During the development of our research (2015), 1074 municipalities were engaged in the program in different stages of implementation with approximately half of them having implemented at least one project funded by the program [21]. Projects e.g., on healthy eating behavior or smoking cessation were implemented in Argentina. The implementation process and the impact of such programs were rarely evaluated [22].

Evaluation of local government interventions is essential for the detection of barriers, justification of expenditures, informed decision-making, and scale up of certain interventions [23,24]. Using knowledge about obstacles and facilitators throughout the projects’ implementation and evaluation as well as scientific evaluation techniques could improve the programs’ effectiveness [25,26,27]. The development of good context specific indicators for impact evaluation is essential for increasing the projects’ acceptability and feasibility. Current research from Latin America provide some information on the implementation and evaluation of local projects of government programs [28,29,30,31,32,33,34,35,36,37]. This research mostly did not explore specific barriers and facilitators in the implementation of the programs. Taking into account that translating the national policy into community level interventions in Argentina might be challenging [35], identifying the barriers and facilitators in the implementation of local health promotion projects in Argentina is needed.

The aim of this study was to identify barriers and facilitators in the implementation of projects oriented to promote physical activity and healthy diet at a municipal or community level, based on experiences of the HCMP in Argentina. We also aimed to explore the ways local teams are evaluating projects.

## 2. Materials and Methods

### 2.1. Study Design and Settings

A mixed methods approach was adopted, complementing qualitative and quantitative approaches to provide an understanding of a phenomenon. The aim of qualitative phase was to develop concepts based on the understanding of the phenomena in natural settings, giving emphasis on the meanings, experiences and views of the participants. That approach gave us knowledge on the phenomenon and allowed us to develop a standardized electronic survey. Qualitative findings were used for developing the electronic survey on domains of the questionnaire, prioritization of questions and response options.

Argentina has 24 provinces that are divided into five geographical regions (Northeast, Northwest, Central region, Cuyo region and Patagonia region). Qualitative interviews were carried out in one or two municipalities of each region. Municipalities were selected in agreement with the HMPC authorities with the intention to cover different types of municipalities regarding population size (including one province capital, and municipalities of different amount of population), urban environment (more and less developed municipalities regarding built environment) and degree of involvement in the program (half of the municipalities with more than two years in the program and some that recently joint). The electronic survey was sent to a broader sample of participants, including all the contacts of communities and municipality members participating in the program in all the provinces. The survey was sent to all participants registered in HMCP database; the total number of recipients is unknown. This study was performed from March to December 2015.

#### 2.1.1. Qualitative interviews

Qualitative research used inductive inquiry consistent with the grounded theory approach [38] and was conducted in accordance with qualitative research guidelines [39,40]. Grounded theory approach was used to inductively generate theory out of the data and its domains. We carried out in depth semi-structured interviews with key-informants purposely selected to cover relevant stakeholders involved in the program as well as group interviews. The group interviews were formed spontaneously by people related to each other by the HMCP. 

We included five target groups: (a) HMCP officers at the national level, who assist in the coordination of activities undertaken in the provinces and support provincial and municipal teams (henceforth referred as “HMCP National referents”); (b) provincial HMCP representatives linked to the provincial governments, who give support to the municipal teams in their projects as well as coordinate activities (henceforth referred as “HMCP Provincial referents”); (c) representatives of the HMCP at the municipality level, i.e., stakeholders responsible for the implementation of the program and follow up of related activities (henceforth referred as “HMCP Local referents”); (d) civil society representatives actively participating in projects that promote healthy habits and taking part in the inter-sectorial committees’ round tables (henceforth referred as “Civil Society members”), and (e) municipal authorities: in two municipalities, we interviewed high-level local authorities, a mayor and a secretary of public health (henceforth referred as “Municipal authorities”). It was not originally planned to include this kind of interviewees but during the fieldwork we considered their input could be meaningful.

Questionnaires for semi-structured interviews included the following themes: experience with implementing projects focusing on promoting physical activity and healthy diet; and key barriers and facilitators on designing, implementing and evaluating projects at the local level. 

A team of researchers (MB, MRC, MS) with experience in qualitative methods conducted the interviews at the municipalities evaluated. A thematic guideline was performed based on a comprehensive review of the HMC initiatives in Latin American countries. Many training sessions were carried out with the team to adjust the domains and the main categories of analysis. We made pilot interviews to test the comprehension and the accuracy of the questions.

All the interviewees gave their verbal informed consent to participate in the study. Most of the interviews were face-to-face and only one was conducted by telephone. Each interview lasted between 20 and 60 min and all the interviews were audiotaped and subsequently transcribed in preparation for the analysis.

#### 2.1.2. Electronic survey

For the electronic survey we used a mailing list provided by the HMCP National referents. The contacts in the list included HMCP Local referents and Civil Society members of inter-sectorial committees. The survey was based on the qualitative findings and a literature search about indicators used in Latin American countries initiatives to assess the performance of the community programs related to the promotion of healthy lifestyles and improvement of the built environment. 

The instrument used in the survey contained both closed multiple response questions and opened questions. Themes included were: socio-demographic data (sex, age and geographical region), experiences in projects related to the promotion of healthy habits during the last 5 years, barriers for projects’ implementation, use of process and outcome measures for projects’ evaluation. The web-based questionnaire was developed using Survey Monkey^®^, which is a web-based, flexible, scalable, secure survey development tool. These platforms led us create questions breaks to avoid queries. We sent the survey to a couple of people to pilot the readiness of questions and the functioning of the platform. Three reminders were sent every 2 weeks to improve the response rate. Finally, a database was created by the platform and downloadable in excel sheets

### 2.2. Data Analysis

The written transcripts from the qualitative interviews were entered into ATLAS.ti version 7 (2013) qualitative data management software (Scientific Software Development, Berlin, Germany). Two researchers (MB and MC) coded the transcripts according to a codebook based on themes included in the questionnaires and supplemented by a grounded theory-based approach to capture emergent themes. First, two transcripts were coded by both researchers in order to agree on the use of the codebook. Thematic analysis was done for each target group. Then, matrices were developed to facilitate comparisons across the transcript materials and to retain the context of the data (i.e., informants target groups and sites). Finally, data was extracted and interpreted. As part of this analysis, direct representative quotations of the participants’ opinions were selected and included in this manuscript to illustrate the findings. In order to protect the identity of the informants we only provide information of type of informant and city (in brackets). Regarding the survey, we conducted a descriptive analysis of continuous variables using the mean or median when appropriate. For categorical variables, absolute frequency and proportions were calculated.

### 2.3. Theoretical Framework

The findings from the qualitative interview provided a large amount of meaningful information regarding the domains explored in the study. These findings helped the researchers to develop a theoretical framework that was used for the survey development and guided the report of the results. 

Figure 1 illustrates the way the results are described. We focused the analysis on the challenges experienced during the implementation and the experiences on evaluating the process or impact of the interventions. The findings describe the barriers and facilitators of the implementation and the measures used for evaluation.

## 3. Results

### 3.1. Characteristic of Qualitative Interview Participants

As shown in Table 1, a total of 44 key informants were interviewed. Six HMCP National Referents participated in two group interviews (2–4 participants each), five were women and all had a university degree in social or medical sciences (sociology, anthropology, psychology and nutrition). We conducted 5 interviews with HMCP Provincial Referents from five provinces each in a different one of the five country geographical regions, three were women and most had a university degree.

For the interviews with HMCP Local Referents we included seven municipalities from the five geographical areas: two municipalities the North West, two in the North East, one in the Centre, one in the West, and one in the South. Four of the interviewees were women and all were older than 39 years. Civil Society members represented 54% of the sample (n = 24), and were recruited from six municipalities in four provinces from four distinct geographical areas (two in the North West, two in the North East, one in the Centre, and one in the West). The age of this target group ranged from 18 to 50 years old, eighteen were women. Finally, we included two local authorities: the mayor and the secretary of health of two different municipalities. 

### 3.2. Characteristics of Electronic Survey Participants

The electronic survey was completed by 206 individuals from at least 96 municipalities. 71% were men, median age 45 years old (range 23 to 70). Most of the respondents (93%) belonged to municipalities participating in the HMCP. Regarding the role of the participants in the program, 2% were HMCP Provincial referents, 55% HMCP Local referents; 22% were Civil Society members participating in the inter-sectorial committees affiliated to the HMCP and 21% did not have a specific role in the program. All the geographical regions were represented with a larger proportion of the central area, which has the largest concentration of population (Table 2).

Most of the survey respondents (91%) reported that their municipalities had implemented at least one project in the last 5 years that aimed to promote physical activity and healthy diet. 

### 3.3. Barriers in the Implementation of Projects Oriented to Promote Healthy Habits at Local Level

Various challenges that hindered the project implementation were identified during the qualitative approach. All the barriers identified during the qualitative approach were included in the survey by asking participants if they had experienced those barriers in the implementation of projects during the last 5 years (Table 3). 

Some of the most important barriers mentioned during the qualitative approach were also frequently selected by the survey respondents. During the interviews, it was mentioned that lack of funds is a problem faced by local initiatives (option selected by 43% of the electronic survey respondents). Many projects require skilled human resources for performing specific tasks. Sustainability of some interventions may be affected by the fact that, usually, appointed personnel work on voluntary basis and therefore fail to maintain commitment over time. The lack of skilled human resources needed to perform tasks was experienced by 42% of the electronic survey respondents.
“For most of the projects we spend the money in communication campaigns. However, you will hardly find a well-designed campaign, because there is a deficiency in professional human resources (…) There is a shared idea that if you worked in communication anyone can do it just as it shall seems it is good” (HMCP Provincial representative)
“We have not in fact had technicians or access to information (…) I must say that I did my greatest efforts to try to understand some facts, but statistics is not for me” (HMCP Local referent)

Lack of material resources (option selected by 31% of the electronic survey respondents) may be due the unrealistic proposed actions for available funds.
“To work without resources is an art. That situation is really an odyssey; you develop a great creativity (HMCP Local Referent)

During the interviews, it was mentioned that the lack of support from the authorities would contribute to the failure of the projects. 17% of the surveyed respondents had faced this problem.
“We know that some mayors are more interested in the partisan politics rather than public politics. We cannot deny that municipalities are political institutions. If the mayor changes, despite any installed capacity, many times the priorities change” (HMCP Local Referent)

Almost 20% of the respondents in the survey identified that the lack of technical support in relation to conducting a project and analyzing and interpreting the data is a barrier experienced in the project implementation. The lack of training in how to design projects was indicated by 12% of the surveyed participants as a hindering factor in implementation.
“It is usual that the proposed projects always include the same activities. I think that it would be useful if we could provide more tools for developing health promotion projects” (HMCP National Referent)

Community acceptability was mentioned as a very important factor to guarantee the success of the projects; therefore, the lack of this factor may be a barrier. Only 9% of the survey respondents mentioned the lack of acceptability in the community as a barrier in the implementation process.
“Regarding the work done to promote family orchards, it was very difficult to get the buy in of the community (…) this program was very hard to settle some years ago” (Civil Society representative)

Most survey respondents mentioned having experienced at least one of the stated barriers while 16% of them indicated that they had not experienced any major difficulties or barriers in implementing projects. 

### 3.4. Facilitators in the Implementation of Projects or Interventions Oriented to Promote Healthy Habits at Local Level

During the qualitative research, we identified some factors that may contribute to the successful implementation of projects undertaken at the local level. 

The **availability of local data about the health situation** was recognized as an important factor for problem prioritization and determination of which interventions are needed. However, the collection and report of data was considered a large challenge at local level. From the interviewees’ point of view, it was difficult to get accurate data. With the support and advice of the HMCP referents, some municipalities developed registers to collect local data. Despite being considered a bottleneck, once this barrier was overtaken, availability of local data was considered one of the main strengths of the HMPC. For some key informants involved in the qualitative approach, this part of the process implied a *“revolution in identifying problems and prioritizing actions” (HMCP National referent).*

In the qualitative research, **inter-sectorial nature of the projects** and **community participation** were considered as key aspects when implementing activities that aimed to improve health. In one HMCP Provincial referent’s own words, this is considered to be very important and expected to *“encourage the articulation among different sectors and to be known by the community, and that the community participates in the project” (HMCP Provincial referent).* In the survey, we asked respondents how often community members are involved in the project’s formulation: 43% of the respondents selected always or almost always, 43% said only sometimes; only 2% responded never. 

According to the interviewed key informants, **previous experience** with working in inter-sectorial teams in order to design project activities facilitated the implementation. 

During the qualitative interviews, participants stated that capacities of **trained personnel, use of local resources** as well as **correct planning of required resources** impacted the implementation of the activities. 

A **motivated and motivating leader** was seen as a mandatory factor for successful projects implementation. According to the key informants, this local individual was proactive with leadership qualities that participated in activities and guaranteed the sustainability of the initiatives. 


*“Our programs usually survive, in quotation marks, because we have a referent that wants to do things well (Provincial referent)*



*We haven’t had provincial support. It was always Bob (fake name for a Local Referent) who pushed things to be done… If it wasn’t for him nothing would be done” (Civil Society representative)*


Successful implementation and continuity of the projects was determined by the *support of the local authorities* and their commitment in implementing the activities. The following quote summarizes some of the facilitating factors mentioned so far: 


*“This has a lot to do with the capacities of the municipality; I mean, if the municipality has a good team and political support; if it makes a difference compared to other municipalities, because they have a team that thought through, that planned things well, that has a strategy of how to do things, then the view on how to implement the project is different and the success is almost guaranteed.” (HMCP Provincial referent)*


Finally, we asked survey respondents to what extent the local projects accomplished the improvement of healthy behaviors: 50% considered very much or quite a lot, 40% somewhat, and 6% nothing.

### 3.5. Experiences in Projects’ Process and Outcome Evaluation

There was a shared perception among interviewees from National, Provincial and Local level about a limited evaluation of the implemented projects. They were not used to formally evaluating results. It was considered difficult to measure long-term effects of small and isolated initiatives implemented at the local level concerning health promotion. Most of the interviewees emphasized that there is a need to improve evaluation. 

According to the interviewed key informants, the use of outcome indicators was less common than the use of process indicators.


*“There are no indicators to measure impact. In general, one of the most important deficiencies we have at health promotion activities is that we only measure process. It is about to register how many workshops we did, how many activities… however they will hardly use measures about the real impact in the population” (HMCP Provincial referent)*


During the inductive qualitative approach, we identified that different measures were used to evaluate the projects. Measures mentioned by the interviewees, including process and outcome evaluation, were included in the survey (Table 4). 

Regarding outcome measures in our survey, 21% of the respondents reported having measured objective health or behavioral outcomes before and after project implementation. For example, clinical parameters like blood pressure or body mass index were measured before and after the intervention at some schools. Health indicators were not only registered by tools especially designed for the project, but also through indicators reported in other surveys (for example the Risk Factor National Survey) [41] which were used to compare parameters before and after project implementation, option selected from 12% of survey respondents. 

Interviewees in the qualitative approach mentioned the use of satisfaction surveys as an outcome. Nearly 14% of the surveyed respondents stated to have implemented these surveys.

The design and implementation of public policies as a product of a project was considered an outcome measure. About 31% of the respondents indicated that the design and implementation of public policies was a measure they used in order to evaluate the outcome. 


*“Local small actions are also having effects regarding the legislation, some regulations were promoted.” (HMCP National Referent)*



*“At our municipality we have promoted municipal regulations. One of the ordinances says that we adhere to healthy kiosks at schools.” (HMCP Local Referent)*


Moreover, in projects that involved the construction of public places fostering physical activity, the ascertainment of characteristics of the environment before and after the intervention had been used as an outcome measure.

The fulfillment of proposed activities was used as an indicator for process evaluation and was mentioned by 53% of the surveyed participants. For example, it was evaluated whether a predefined number of attendants to educational workshops or a predefined number of schools implementing healthy snacks kiosks was reached or not. These measures represented a simple and low-threshold way to evaluate predefined goals. 


*“Let’s say it was a workshop […] for adolescents […], and you already did four workshops with 20 adolescents. You have a goal, you set a predefined goal; there were 10 people in one workshop, 5 in the other, and so on. So […] there is the result I expect, the predefined goal, and then there is the result I obtain. And that is how they evaluate, with these indicators.” (HMCP Provincial referent)*


Another process indicator mentioned by the interviewees was the total number of people reached by the project intervention over the total of potential beneficiaries. This was reported by 23% of the survey respondents.

## 4. Discussion

This study provides an important insight on how local initiatives that promote healthy behaviors are implemented and evaluated in Argentina. This study revealed several barriers hindering successful implementation and, above all, revealed the need for process and impact evaluation. 

The most frequent barriers experienced at local level were the limited financial resources, lack of material resources and lack of skilled human resources. Less frequent but nevertheless important were lack of technical support to design or evaluate projects; lack of training on project design; and lack of support from local authorities or acceptance of the proposed intervention by the local community. 

In addition to identify perceived barriers in implementing local projects, this study gave us significant information about positive experiences in the implementation process. One important finding is that a motivated local leader, who coordinates activities and encourages the project team, is perceived as a key condition for success. It is interesting to note that local projects were widely accepted and supported by local authorities and by the community. Moreover, municipalities that have support from local authorities and involvement of community members, effectively promoting inter-sectorial participation, seem to have more chances to implement and sustain initiatives. Interventions on the built environment are a key opportunity to influence physical activity and diet of the population through health promotion initiatives. [16,17]. The availability of accurate data is critically important in order to prioritize the problems to be approached. In contrast with the lack of resources mentioned as a barrier, initiatives planned by seizing local resources and capitalizing previous experiences represent a facilitator.

In accordance with other authors, we found that the evaluation of local projects was mostly based on process rather than outcome indicators [28,42]. According to our findings, register of the fulfilment of proposed activities are process measures frequently used. However, there are some experiences that use impact evaluation, such as objective measures before and after the project implementation, as well as user satisfaction surveys. 

The HMCP was implemented in many countries of Latin America. Like in Argentina, the majority of the implemented projects aimed to promote physical activity and healthy diet through lifestyle change interventions and healthy environment initiatives. Similar projects were conducted in other countries, for example in Brazil, Paraguay, Guatemala, Ecuador, Bolivia and Chile [29,30,31,32,33,34]. Due to the similar nature of projects implemented across countries, shared facilitators and barriers of the implementation and evaluation process could be comparable to those we found in Argentina. However, different organizational, financial and personnel preconditions in each country may facilitate or hinder implementation and lead to different experiences.

Our results are in accordance with the survey conducted by Meresman et al. [35] in 2008 when the HMCP was young in Latin America. At the time, they voiced the need for citizen empowerment and the need to monitor and assess the impact of the programs on health and quality of life of the populations involved. Merseman et al. reported a lack of training regarding management, program planning, and evaluation of the impact of HMCP projects, which is in accordance with our study results. Additionally, existing training is usually not maintained in most Latin American countries. Tutoring and forums promoting exchange and reflection among peers might provide the opportunity to learn from the others experiences without being too resource-intensive. 

In other Latin American countries evaluation also commonly relies on process indicators, for example, whether a required number of scheduled activities was reached or not [29,30,31,32,33,34,36,37,43,44,45]. Other authors also stated that challenges included how to secure resources for the evaluation, time constraints, and working in an inter-sectorial manner [28]. Some initiatives addressed different methods of community participation (forums, open hearing and participation maps). For example, the Bertioga Municipality Healthy Project prioritized interventions using action research approaches and participatory techniques, aiming at the mobilization of social actors and the creation of collective spaces for discussion and promotion the project in the many regions of the municipality.

Project evaluation that includes not only process indicators, but also outcome measures, is essential in order to decide on project’s continuance, derogation or adaption, and in order to ensure appropriate resource allocation. The evaluation of outcome measures also allows for determining the effect of the projects on the health of the population. Hence, the importance of outcome measures should be more stressed in the evaluation of the projects [27]. We consider that the same barriers and facilitators identified for the implementation stage may also be present during the prioritization of the problem, selection of the intervention and throughout the evaluation process.

We believe that there is a need to develop indicators for project evaluation that are feasible to measure in local communities, and to train local human resources on this activity. We emphasize that indicators and strategies for project evaluation should be culturally adapted taking into account local capacities and experiences. We encourage communities and municipalities developing local registries with reliable data and strengthening their surveillance system. 

### Strengths and Limitations

The mix-methods approach allowed us to describe a broad picture of the stakeholders’ experiences in the implementation and evaluation of health promotion projects by triangulating qualitative and quantitative sources. The in-depth semi-structured interviews gave us detailed descriptions of the barriers and facilitators from the stakeholder’s perspective, while the survey enabled us to examine how often these barriers and facilitators were present. One limitation of the methods is that findings were based on opinions and self-reported rather than on observation of the projects. However, the sample included a wide range of stakeholders and we found that the opinions were consistent. We could only collect few characteristics of the surveyed respondents to preserve data confidentiality.

We are aware of a potential social desirability bias (the wish to appear as a morally-worthy person to the interviewer) from the qualitative interviews. In order to minimize this bias in sensitive questions, as for example when asking about the quality of the work they do, where people may feel judged, we avoided leading questions allowing respondents to express freely. 

The survey was sent by e-mail to all participants registered as contacts in the inter-sectorial committees of the affiliated municipalities from the HMCP. We are aware that email-administered surveys may have some limitations as representatives at local level might not have been reached due to lack of internet in some areas. Additionally, there might be a selection bias as only the most motivated and committed representatives might have completed the survey. The most important limitation we faced is that the total number of valid email addresses in the registry is unknown and therefore the response rate of the survey could not be calculated. Despite this limitation, we obtained information from a big range of geographical regions and the sample size covered the main actors involved into the HMCP. 

## 5. Conclusions

This study contributes to better understand difficulties in the implementation of community-based intervention projects. It also contributes to identify factors that enable successful implementation of initiatives aimed to promote healthy habits. The findings could inform stakeholders about barriers they may face and could guide them on how to facilitate local initiatives when implementing a project. There is a need to improve project evaluation strategies by incorporating process, outcomes and context specific indicators. 

The mixed-methods approach contributes to describe a broad picture of a complex process that has been little described in the literature. The study capitalized on the Argentinean HMCP initiative as a source of institutional knowledge. Social Ecological Model (SEM) based investigations should be developed for understanding the multifaceted and interactive effects of personal and environmental factors that determine behaviors.

## Figures and Tables

**Figure 1 ijerph-16-00213-f001:**
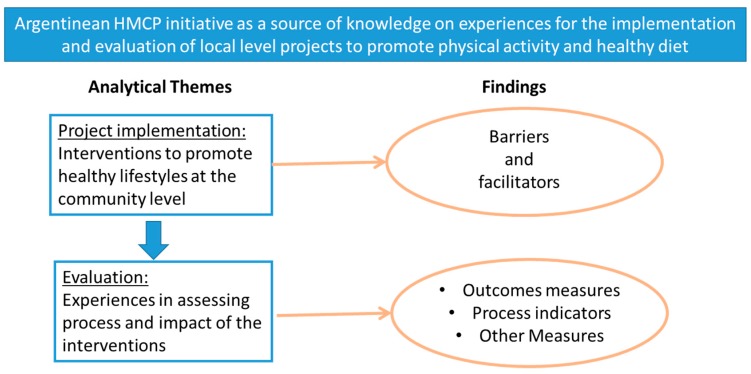
Theoretical framework.

**Table 1 ijerph-16-00213-t001:** Characteristics of the participants in qualitative interviews (n = 44).

Characteristics	n	(%)
Target group		
HMCP National Referents	6	13.6
HMCP Provincial Referents	5	11.4
HMCP Local Referents	7	15.9
Civil Society Representatives	24	54.5
Municipal Authorities	2	4.5
Geographical area		
North West	8	18.2
North East	18	40.9
Centre	6	13.6
West	8	18.2
South	3	6.8
Unknown	1	2.3
Gender		
Female	30	68.2
Male	14	31.8

**Table 2 ijerph-16-00213-t002:** Characteristics of respondents of the electronic survey (n = 206).

Characteristics	n	(%)
Role in the HMCP		
Provincial representatives	4	2.0
Municipal representatives	95	46.6
Members of inter-sectorial round table	38	18.6
No active role in the program	36	17.6
Missing	332	15.7
Geographical area		
North West	14	6.9
North East	14	6.9
Centre	32	15.7
West	14	6.9
South	14	6.9
Missing	118	57.4
Gender		
Female	71	34.8
Male	57	27.9
Missing	78	38.2

**Table 3 ijerph-16-00213-t003:** Barriers for the implementation of local projects aimed to promote healthy habits from survey respondents (n = 138).

Barriers	n	%
Lack of adequate funds	59	42.8
Lack of skilled human resources for performing tasks	58	42.0
Lack of materials to perform activities	43	31.2
Lack of technical support to conduct, analyze and interpret the information	28	20.3
Lack of local authorities support	24	17.4
Lack of acceptance from the community	13	9.4
Lack of training on how to design a project	17	12.3
Other barriers	3	2.2
We have not had projects with major difficulties	22	15.9
Don’t know	9	6.5

Because more than one answer was possible, the total sum of responses exceeds n = 138 and 100%.

**Table 4 ijerph-16-00213-t004:** Outcomes and process indicators used in local projects from survey respondents (n = 145).

Type of measuring data	n	%
*Outcome measures*		
Public policies designed and implemented (e.g., Municipal ordinance or other regulation)	45	31.0%
Objective measures before and after the implementation of a project (e.g., weight and height, cholesterol measures, questionnaire on exercise and dietary habits)	30	20.7%
User’s satisfaction surveys	20	13.8%
Health indicators reported in other surveys (Risk Factor National Survey, provincial statistics or hospital records, etc)	18	12.4%
*Process indicators*		
Proposed activities fulfilment (e.g., number of attendants to educational activities as workshops or seminars, number of schools that implemented healthy snacks kiosks, number of persons that used facilities)	77	53.1%
Total number of people reached by the project intervention over total of potential beneficiaries	33	22.8%
*Other Measures*		
Number of potential beneficiaries	46	31.7%
Records of social workers or community health workers	14	9.7%
Achievements were not measured	23	15.9%
Don’t know	9	6.2%

Because multiple responses were possible, the total sum of responses exceeds n = 145 and 100%.

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
