# Peer review of "Barriers and Facilitators for the Implementation and Evaluation of Community-Based Interventions to Promote Physical Activity and Healthy Diet: A Mixed Methods Study in Argentina"

_ijerph, 2019, doi:10.3390/ijerph16020213_

Round 1
Reviewer 1 Report
See attached comments and suggestions.

Author Response
Response to the Reviewer#1
Thank you for the comments and suggestions. We appreciate the summary you developed from our study.
We gave numbers to your comments in order to respond to each of them.
Comment #1
Introduction: The introduction to this paper provides a compelling case for the need to better understand potential determinants of success in implementing and evaluating lifestyle interventions.
Authors´ Response:
Thank you for your positive comments. We have made few editions requested by other reviewers.
Comment #2
Methods: Some key information is missing. For example, much more is needed on the researchers' bases and process for selection of participants in the interviews. This would help the reader to better assess the representativeness and potential sources of biasing in the sample across the participant domains. To give just two examples: How and to what extent was geographical sampling /representation used?
And why were only two, and these particular two, high-level municipal authorities recruited for participation.
It should also be noted in the methods section that inconsistencies in the email list for the electronic surveys precluded an accurate assessment of percentage of participation in the survey. This is mentioned in the discussion section, but the reader will want that information in the methods section.
Clarity is also needed regarding the electronic survey on whether participant representation by region, for example, overrepresentation in the central region, occurred naturally, or was purposeful as part of the study design.
In addition, no mention is made of the selection, number, background, training, supervision, and evaluation of fidelity of the coding and other evaluation personnel. Also, who conducted the semi structured interviews, and what steps were taken to assess and ensure fidelity to the process?
Authors´ response:
We added more details regarding the process of selection of participants interviewed and criteria for selection.
We added a statement explaining that it was not plan to include high level municipal authorities, but during the field work we considered their input would be meaningful.
We added a statement in the methods section about the fact we couldn´t know the total number of recipients of the email list.
We are including more detail/clarity regarding the representation of geographical areas; the data collection process, and coding for the analysis.
Comment #3
Results: Results describing barriers and facilitators at the local level are addressed adequately and well supported by illustrative quotes. Results on evaluation appear to be most informative as to the inconsistencies - and perhaps lack of knowledge or guidance? - in conducting outcome measurement (along with meaningful process measurement). It would be helpful to know what guidance exists in the HMCP program to train, consult and supervise local authorities to conduct evaluations, particularly outcome evaluations.
Authors´ response:
Thank you for your suggestion. We have added the information regarding the guideline promoted by the program to conduct the evaluations. PAHO Guideline (Healthy Municipalities & Communities: Mayors’ Guide for Promoting Quality of life)
Comment #4
Discussion: The importance of outcome measurement in the future is discussed as a problem in this section, but no detailed ideas are given on how to address: (1) the lack of knowledge and skills of local personnel for undertaking reliable and valid outcome evaluations; and (2) resources for consultation and supervision on development and implementation of outcome evaluation, particularly if it is to be culturally adapted by region/locality, as the authors suggest.
Authors´ response:
Some ideas based on other countries experiences were added and have added some suggestions on how to address the lack of training regarding evaluation.
Reviewer 2 Report
Introduction: could be summarised for brevity.
Methods: could provide more specific and why each method is appropriate. Who provided ethics approval.
Results: Provide clear profile of the respodents, Could tabulate the results and comments to show common themes. Some statistics could help summarise outcome measures.
Author Response
Thank you for the comments and suggestions. We gave numbers to your comments in order to respond to each of them.
Comment #1
Introduction: could be summarised for brevity.
Authors´ response:
We have summarized some paragraph in order to avoid redundancies.
Comment #2
Methods: could provide more specific and why each method is appropriate.
Who provided ethics approval.
Authors´ response:
We have added more detail in the methods regarding why each method is appropriate
We have completed the information about the ethical committee that approved the study in a specific section requested by the journal. We wonder if we need to add that information also in the body of the text. We kindly ask the editors to advise on this specific issue.
Comment #3
Results: Provide clear profile of the respodents, Could tabulate the results and comments to show common themes. Some statistics could help summarise outcome measures
Authors´ response:
We included all the information we collected regarding the respondents profile.
Regarding the statistics requested to summaries outcome measures, we are not able to perform that kind of analysis. We have added that in the limitation section.
Reviewer 3 Report
This paper is interesting and I recommend that it is considered for publication: it is clear that a considerable amount of work was conducted by the researchers and the paper deals with an important area of understanding the difficulties of implementing community-based intervention projects. I hope the below comments might be of use to the authors and allow them a little more time to develop/present the framing and interpretation of their results.
Introduction
The introduction and discussion this could be more concise, which would allow the authors to add in a couple of extra paragraphs to address other points that would aid the background/discussion argument. For example, the reference to individual behaviour and systems/environments was excellent (lines 39-41), perhaps this could be reflected upon in the discussion, especially in relation to the 'inter-sectorial nature of the projects' finding?
Had the authors considered providing a brief overview of the types of interventions that were being evaluated/implemented and previous research in this area i.e., to emphasise what is new/different about Argentina and how the results of this study might aid discussions around intervention implementation at a community level (e.g., are there any particular challenges in translating national or international policy into community level interventions in Argentina)?
Line 47 'This problem'. I would suggest this paragraph is rephrased, as it might be unclear to the reader what 'this' is referring to. Perhaps begin the paragraph with WHO Ottawa Charter and then the regional activities in Americas (HMCP) to clarify their relationship.
Methods/results/discussion
Please clarify the qualitative approach taken e.g., further detail on the use of grounded theory (continued comparison/theoretical constructs), or perhaps consider if thematic frameworks were employed. Also, provide a sentence more detail on line 87 i.e., so the reader can understand what was followed/considered important from Mays et al etc. Natural groups requires clarification.
It would be a huge asset to this paper if the authors were able to provide a small amount of detail on how one used the qualitative data to inform the survey data: in what way was the survey 'based on the qualitative findings' (response options, framework/domains of questionnaire, theoretical constructs)?
Consider whether the results section is presented in the best way to show the large amount of work conducted and the findings e.g.,. are these results presented in the framework of analysis? clarify the main findings to be discussed. Also, double check the numbers in the tables add up to the totals (100% or n=206). Would it be useful for data interpretation to present information on the types of municipalities (pop size, urban environment, degree of involvement in programme)? What were the response rates, is this a limitation?
The co-production/community involvement aspect of this work can be embeded in a lot of existing research. If this is one of your main interesting findings you might like to discuss this a little more (and add some references in the intro and/or discussion).
Do your results suggest training is required to evaluate project outcomes, if so what would that look like and how might it be actioned (bearing in mind the financial and material resources also mentioned)? Are their any international examples of how this is being done currently?
Conclusion
A lot of work has been done and I believe this conclusion could be stronger/clearer if the authors have the time to a take a little longer reflecting on what was done and how it contributes to the existing research.
Throughout
A minor thing, I would rephrase certain sentences/avoid colloquial or emotive words throughout e.g., about (line 59), remarkable (line 316), better (line 329); omit the use of however if it is not required; and run a typo check (e.g., 'obtainment' line 358).
Good luck with the revisions and I wish you all the best!
Author Response
Thank you for the comments and suggestions. We appreciate your advice about making more visible the amount of work conducted.
We gave numbers to your comments in order to respond to each of them.
Comment #1
Introduction
The introduction and discussion this could be more concise, which would allow the authors to add in a couple of extra paragraphs to address other points that would aid the background/discussion argument. For example, the reference to individual behaviour and systems/environments was excellent (lines 39-41), perhaps this could be reflected upon in the discussion, especially in relation to the 'inter-sectorial nature of the projects' finding?
Authors´ response:
We have summarized some paragraph in order to avoid redundancies.
We have added in the discussion a comment about the advantages of healthy environments.
Comment #2
Had the authors considered providing a brief overview of the types of interventions that were being evaluated/implemented and previous research in this area i.e., to emphasise what is new/different about Argentina and how the results of this study might aid discussions around intervention implementation at a community level (e.g., are there any particular challenges in translating national or international policy into community level interventions in Argentina)?
Authors’ response
We have included some information on projects in Argentina and other Latin American countries and also provided information on the lack of current research regarding experiences with barriers and facilitators in the implementation of the programs in Latin America, especially in Argentina.
Comment #3
Line 47 'This problem'. I would suggest this paragraph is rephrased, as it might be unclear to the reader what 'this' is referring to. Perhaps begin the paragraph with WHO Ottawa Charter and then the regional activities in Americas (HMCP) to clarify their relationship.
Authors´ Response:
Paragraph was edited as suggested
Comment #4
Please clarify the qualitative approach taken e.g., further detail on the use of grounded theory (continued comparison/theoretical constructs), or perhaps consider if thematic frameworks were employed. Also, provide a sentence more detail on line 87 i.e., so the reader can understand what was followed/considered important from Mays et al etc. Natural groups requires clarification.
Authors´ Response:
We edited the methods section in order to include the information suggested
Comment #5
It would be a huge asset to this paper if the authors were able to provide a small amount of detail on how one used the qualitative data to inform the survey data: in what way was the survey 'based on the qualitative findings' (response options, framework/domains of questionnaire, theoretical constructs)?
Authors´ Response:
We added some statements to explain how the qualitative data was used to develop the survey. Moreover we included a brief section within the methods for the Theoretical framework.
Comment #6
Consider whether the results section is presented in the best way to show the large amount of work conducted and the findings e.g.,. are these results presented in the framework of analysis? clarify the main findings to be discussed. Also, double check the numbers in the tables add up to the totals (100% or n=206). Would it be useful for data interpretation to present information on the types of municipalities (pop size, urban environment, degree of involvement in programme)? What were the response rates, is this a limitation?
Authors´ response:
In order to explain the way the results are presented and to help the readers to follow the manuscript, we are including in this new version a brief section were we describe the theoretical framework. We hope this clarify the way the results are presented and also.
We have checked the numbers in Table 1 and Table 2, so that the totals add up to 100 % and the total n.
We donot have the data about type of municipalities of the survey participants. We added a statement of the lack of that information in the limitations. We have not the total number of survey recipients and that is also detailed as a limitation.
Comment #7
The co-production/community involvement aspect of this work can be embeded in a lot of existing research. If this is one of your main interesting findings you might like to discuss this a little more (and add some references in the intro and/or discussion).
Authors´ response:
We added more discussion on existing research.
Comment #8
Do your results suggest training is required to evaluate project outcomes, if so what would that look like and how might it be actioned (bearing in mind the financial and material resources also mentioned)? Are their any international examples of how this is being done currently?
Authors´ response:
We have added some examples on how training in evaluation could be implemented. We have stated, that the training is weak in other Latin American countries as well.
Comment #9
Conclusion
A lot of work has been done and I believe this conclusion could be stronger/clearer if the authors have the time to a take a little longer reflecting on what was done and how it contributes to the existing research.
Authors´ response:
Thank you for your suggestion. We have edited emphasizing the importance of the findings.
Comment #10
Throughout
A minor thing, I would rephrase certain sentences/avoid colloquial or emotive words throughout e.g., about (line 59), remarkable (line 316), better (line 329); omit the use of however if it is not required; and run a typo check (e.g., 'obtainment' line 358)
Authors´ response:
Thank you. We edited as suggested